# Systemic Delivery Strategies for Oncolytic Viruses: Advancing Targeted and Efficient Tumor Therapy

**DOI:** 10.3390/ijms26146900

**Published:** 2025-07-18

**Authors:** Yunxin Xia, Dan Li, Kai Yang, Xia Ou

**Affiliations:** School of Medicine, Kunming University of Science and Technology, No. 727, Jinming South Road, Chenggong, Kunming 650500, China; prience81@126.com (Y.X.); bingbingwz@126.com (D.L.); heyou12@126.com (K.Y.)

**Keywords:** oncolytic virus, intravenous injection, delivery methods

## Abstract

The rapid development of therapies using oncolytic viruses (OVs) has highlighted their unique advantages, such as their selective replication in tumor cells and their activation of a specific systemic antitumor immune response. However, effectively delivering OVs to tumor sites, especially solid tumor sites, remains a critical challenge. Intratumoral injections face significant barriers in treating some malignant tumors in internal organs, while increasing preclinical data support the use of intravenous injections. Nevertheless, intravenously injected viral particles may be prematurely cleared by circulating antibodies or complements, resulting in a reduced virus dose effectively reaching the tumor site. Therefore, developing methods to shield viruses from the neutralizing environment of the bloodstream while heading toward tumor sites is a must. In this review, we discuss some of the most promising delivery methods for OVs currently under investigation.

## 1. Introduction

OV therapy for cancer holds significant advantages [1]. Compared to traditional treatments, including radiotherapy, chemotherapy, and surgery, the primary advantage of OVs lies in their selective replication within tumors. This selectivity is mainly attributed to two factors: First, OVs can selectively bind to and infect tumor cells by targeting specific receptors that are highly expressed on the tumor cell surface [2]. For example, Coxsackievirus CVA21 infects tumor cells by binding to intercellular adhesion molecule-1 (ICAM-1) and decay-accelerating factor, both of which are overexpressed on tumor cells. Second, tumor cells often exhibit abnormalities in pathways such as the interferon (IFN), P53, and rat sarcoma/rapidly accelerated fibrosarcoma/mitogen-activated protein kinase kinase/extracellular signal-regulated kinase (RAS/RAF/MEK/ERK) pathways, which facilitate viral survival and replication. In contrast, when OVs infect normal cells, type I IFN production is promoted, and Toll-like receptors are activated, leading to the activation of protein kinase R (PKR). Phosphorylated PKR inhibits viral protein synthesis, thereby blocking viral replication [3]. The second major advantage of OVs is their ability to effectively activate immune responses [4]. Tumor cells are lysed by OVs, leading to the release of tumor-associated antigens (TAAs) and tumor neoantigens (TNAs), which activate specific immune responses. Additionally, OVs can induce immunogenic cell death (ICD), leading to the release of damage-associated molecular patterns (DAMPs), while the viral components of OVs trigger the production of pathogen-associated molecular patterns (PAMPs). The host immune system is activated by both DAMPs and PAMPs. Moreover, the infection of tumor cells by OVs can induce the release of cytokines and chemokines, thereby recruiting and activating additional immune cells.

Despite the aforementioned advantages of OVs, numerous limitations are associated with their current applications. The method of administration is a key factor affecting their therapeutic efficacy. Intratumoral injection is the most commonly used delivery method for OVs in both preclinical studies and clinical applications. This method allows for the precise control of OV concentrations in the tumor microenvironment (TME), thereby achieving better therapeutic outcomes [5]. However, some clinical trials have shown that, due to the dense structure of tumor tissue and the high interstitial pressure in tumors, OVs exhibit poor diffusion in tumor tissues after administration, and their antitumor efficacy is seriously affected [6,7,8]. In addition, it is difficult to treat tumors deep within tissues and organs through intratumoral injection [9], and it must be supplemented with ultrasound, computed tomography, nuclear magnetic resonance imaging, and other technologies to guide OV delivery. Meanwhile, when using this delivery method, there are extreme differences in its effectiveness in treating metastatic tumors and inaccessible tumors, such as those in the brain [10]. The intravenous administration of OVs is a rational way to treat tumors originating in deep organs, recurrent tumors, and metastatic tumors because it can deliver OVs to all parts of the body, thereby improving antimetastasis and antirecurrence abilities [11]. However, many obstacles to clinical application remain: (1) Once exposed, OVs enter the blood circulation, and the body’s antiviral immune response is activated to clear the viral particles in the blood [12,13]. The viruses can be swallowed by antigen-presenting cells (APCs), eliminated by preexisting blood factors (such as the coagulation factors factor IX (FIX), factor X (FX), and complement protein C4b-binding protein (C4BP)) [14], and neutralized by preexisting antibodies [15], which makes it difficult for them to effectively reach target tumors. This result has been observed with various OVs, including herpes simplex virus (HSV-1) [16] and adenoviruses (Advs) in preclinical models. (2) Previous research has shown that viruses easily bind to natural immunoglobulin M (IgM) and activate the complement cascade response, leading to their sequestration in liver Kupffer cells and splenic macrophages by complement- and IgM-dependent antiviral mechanisms [17,18,19]. (3) The intravenous administration of OVs increases the opportunity to make contact with normal tissues, resulting in the non-specific entry of OVs into normal tissues, which may lead to extremely severe toxicity and side effects [20,21]. However, once we can overcome these fatal drawbacks of the intravenous delivery of OVs, it will remain the most promising delivery method. Therefore, extending the circulation times of the viruses in blood and improving viral tropism are key issues in this treatment. There is an urgent need to develop a method that can effectively deliver OVs via intravenous injection. In this review, several effective delivery strategies for OVs are summarized, including cell carriers, protein corona, key capsid protein modification, and nanoparticle (NP) carriers (Figure 1).

## 2. Cell-Mediated OV Systemic Delivery

Cell-based carriers are formed by wrapping OVs with cells or cellular components such as cell membranes to improve circulation times, targeting, and biocompatibility and to decrease antiviral immunity in vivo. The susceptibility of viruses to vector cells, the kinetics of viral replication and release within the vector cell type, the kinetics of vector cell transport from the bloodstream to the tumor site, and tropism toward tumors must be considered [22,23]. The following discussion focuses on some of the promising candidate carrier cells that are being developed (Figure 2).

### 2.1. Tumor Cells

During cancer progression, cancer cells may spread to other parts of the body through the bloodstream or lymphatic system. Interestingly, some studies have suggested that, even after cancer cells have disseminated to distant sites, they may retain the ability to return to the primary tumor. This phenomenon is known as “homing”. “Homing” appears to be associated with the same receptor molecules involved in metastasis, such as cell adhesion molecules, chemokine receptors, and integrin ligands [24,25]. Tianyuan Ci et al. discovered that the bone marrow-homing and retention capabilities of leukemia cells are related to the expression of the chemokine receptors C-X-C chemokine receptor type 4 (CXCR4) and cluster of differentiation 44 (CD44) on their cell membranes [26] (Figure 2A). Kefah Mokbel suggested that the circulating tumor cells (CTCs) of breast cancer present in the peripheral circulation possess the ability to home to specific sites. CTCs exhibit the overexpression of CXCR4, while chemokine stromal cell-derived factor-1 (SDF-1, also known as CXCL12), which recruits and retains CXCR4+ cells, shows significant upregulation at the primary tumor site. Parkins et al. successfully detected systemically administered iron-labeled CTCs using magnetic particle imaging (MPI) and visualized tumor self-homing in a human breast cancer mouse model [27]. Although tumor cells exhibit a high capacity for tumor homing, their innate tumorigenic potential raises safety concerns. Therefore, using tumor cells as carriers of OVs often requires inactivating or attenuating them first. Qing Wu et al. developed a cryo-shocked cellular vector for the systemic administration of oncolytic adenovirus type 11 (Ad11), which was internalized into tumor cells via cluster of differentiation 46 (CD46)-mediated endocytosis. After liquid nitrogen treatment (LNT), the pathogenicity of the tumor cells was eliminated. When exposed to the bloodstream, the ligands and receptors on the surface of tumor cells, including CD44 and E-cadherin, along with the enhanced retention in the pulmonary capillaries due to their micron-sized dimensions, facilitated the accumulation of LNT-Ad11 in lung metastatic lesions. This resulted in the suppression of tumor progression and an increasing infiltration of CD8^+^ and CD4^+^ T cells in the tumor tissue [28].

### 2.2. Monocytes/Macrophages

Monocytes/macrophages can sense cancer-related chemokine and cytokine signals, actively migrate to tumor tissues, overcome various biological barriers, and evade undesired immune responses [29]. Studies have shown that macrophages can specifically target tumor tissues through the α4β1 integrin, binding to vascular cell adhesion molecule-1 (VCAM-1) on cancer cells [30]. In addition, chemokines secreted at necrotic tumor sites by C-C motif chemokine ligand 2 (CCL2) and C-C motif chemokine ligand 5 (CCL5) allow for higher macrophage recruitment and infiltration in tumor tissues (Figure 2B). Maria Bunuales et al. demonstrated that, as the most commonly used oncolytic virus prototype, adenovirus type 5 (Ad5) infects human acute monocytic leukemia (MM6) cells, after which these MM6 cells remain capable of sensing CCL2 and CCL5 [31]. Furthermore, preclinical studies have found that macrophages not only uptake and deliver the oncolytic virus HSV1716 to tumors but also support HSV1716 replication within macrophages, potentially enhancing the efficacy of viral therapy [32]. Robert A et al. demonstrated that human monocytes loaded with reovirus can deliver replicative reovirus to tumor tissues, leading to tumor cell infection and lysis [33].

Tumor-associated macrophages (TAMs) are a subset of macrophages that are repolarized in the TME [34]. Compared to conventional macrophages, TAMs demonstrate enhanced tumor infiltration capabilities driven by the tumor microenvironment. This is primarily due to TAMs secreting angiogenic factors, such as vascular endothelial growth factor (VEGF), which promote the formation of new blood vessels in tumors. These newly formed vessels create pathways for the further infiltration of immune cells such as TAMs [35,36]. SANTOS et al. found a significant infiltration of CD68^+^CD163^+^S100- macrophages in tissue sections of plasmacytomas in multiple myeloma patients using immunostaining. CD68^+^ TAMs accounted for 2–12% of nucleated cells, distributed evenly within the parenchymal tissue [37], demonstrating the effective infiltration of macrophages into tumor tissues. In another study, macrophages were cotransduced with a hypoxia-regulated E1A/B construction and an E1A-dependent oncolytic Adv proliferation was also restricted to prostate tumor cells by prostate-specific promoter elements from the tumor-associated receptor protein (TARP), prostate-specific antigen (PSA), and prostate-specific membrane antigen (PSMA) genes. Once these cotransduced macrophages reached hypoxic tumor areas, the E1A/B proteins expressed by the macrophages activated Adv replication, resulting in a significant inhibition of both the growth of primary tumors and the formation of pulmonary metastases [38].

### 2.3. T Lymphocytes

#### 2.3.1. Background

T lymphocytes can freely circulate in the bloodstream and target tumor tissues while also providing synergistic therapeutic effects through their cytotoxic functions [39]. Additionally, chemokines, such as the C-X-C motif chemokine ligand 9 (CXCL9), C-X-C motif chemokine ligand 10 (CXCL10), C-X-C motif chemokine ligand 11 (CXCL11), and C-X-C motif chemokine ligand 12 (CXCL12), secreted in tumors can bind to receptors on the surface of T lymphocytes and initiate downstream signaling pathways. These chemokine receptors are predominantly G-protein-coupled receptors (GPCRs). Upon the binding of chemokines to GPCRs, GDP at the nucleotide-binding site of the Gα subunit is converted to GTP, activating the heterotrimeric G protein, which then dissociates into Gα and Gβγ subunits. The Gα subunit subsequently activates downstream calcium, ERK, and protein kinase B (AKT) signaling pathways, facilitating the migration of T lymphocytes to the tumor site [40] (Figure 2C). Because of these characteristics, T cells can serve as ideal delivery vehicles for OVs. Several different subtypes of T lymphocytes have been studied as cell carriers, including tumor-infiltrating lymphocytes (TILs) [41,42,43], cytokine-induced killer (CIK) cells [44] and chimeric antigen receptor T cells (CAR T cells).

#### 2.3.2. Application of T Lymphocytes as OV Carriers

##### Tumor-Infiltrating Lymphocytes (TILs)

TILs are immune cells present in the TME, mainly including cytotoxic, helper, and regulatory T cells. Patients who have TILs in the stroma may have a better response to chemotherapy and a favorable long-term prognosis [45]. However, the TME is typically filled with immunosuppressive factors, such as transforming growth factor-beta (TGF-β), interleukin-10 (IL-10), and programmed death-ligand 1 (PD-L1), which inhibit the activity of TILs [46]. Nevertheless, some studies have found that the cytokine distribution in the TME can be altered by the infection and destruction of tumor cells by OVs, reducing the effects of immunosuppressive factors and relieving the inhibition of TILs [47]. Simultaneously, TILs, as carriers of OVs, can not only directly participate in the immune attack on tumors but can also help shield OVs from early detection and clearance by the immune system. This allows for the efficient delivery of the virus directly to the tumor site, enhancing the oncolytic activity of the virus and facilitating a dual mechanism of tumor destruction. Consequently, an increasing number of studies are focusing on utilizing TILs as delivery vehicles for OVs. Santos et al. designed TILT-123, an oncolytic Adv expressing TNFα and interleukin-2 (IL-2) in human and hamster tumors. As anticipated, hamster TILs infected by TILT-123 could deliver TILT-123 to tumors through intravenous injection, further inducing the infiltration of TILs containing CD4^+^ and CD8^+^ T cells. It was found that TILT-123 worked synergistically with T-lymphocyte therapy, and 100% of the animals were cured [41]. Additionally, Yuan Ping’s team developed a delivery technology called ONCOTECH (oncolytic virus–T-cell chimera). This approach involves constructing an oncolytic Adv encoding the CRISPR-associated protein 9 (Cas9) gene-editing system (termed eOA), followed by coating eOA with tumor cell membranes expressing ovalbumin (OVA) (termed M@eOA). OVA-targeted CD8^+^ T cells derived from OT-1 mice are then used to carry the membrane-coated M@eOA (termed T-M@eOA). This anchoring strategy does not impair T-cell function, and once the T cells carrying the oncolytic virus reach the tumor cells and recognize their specific antigens, the oncolytic Adv is released and infects the tumor cells. More importantly, both oncolytic virus therapy and adoptive T-cell therapy often lead to high PD-L1 expression in solid tumors, allowing them to evade immune surveillance and resist further T-cell-mediated killing. The Cas9 editor carried by the gene-edited oncolytic Adv can target and knock out the PD-L1 gene in tumor cells and tumor-infiltrating immunosuppressive cells, reducing their PD-L1 expression levels. This reverses the immunosuppressive effects of the tumor microenvironment, a strategy referred to as first-generation ONCOTECH technology [48].

##### Cytokine-Induced Killer (CIK) Cells

CIK cells are a heterogeneous population of polyclonal CD3^+^CD56^+^ T lymphocytes with the phenotypic and functional properties of NK cells, and they are obtained by culturing human peripheral blood single nucleated cells in vitro with a variety of cytokines, including anti-CD3 monoclonal antibody (CD3McAb), IL-2, IFN-γ, and interleukin-1 alpha (IL-1α) [49]. CIK cell cytotoxicity against tumors is exerted in a major histocompatibility complex (MHC)-unrestricted manner through the engagement of natural killer group 2 member D (NKG2D) molecules. Additionally, being independent of traditional cytokine storms and exhibiting lower toxicity, CIK cells offer greater safety when used as a delivery vehicle for OVs, potentially reducing systemic side effects during oncolytic virus therapy. [50]. At the same time, CIK cells can effectively penetrate tumor tissues and deliver OVs to the tumor, which aids in increasing the viruses’ diffusion and infectivity in the tumor, thereby enhancing their antitumor efficacy. Matthias Edinge et al. found that CIK cells were mainly concentrated at the tumor site 72h after intravenous injection and had a better tumor-targeting ability [51]. Steve H et al. knocked out the oncolytic vaccinia virus (VV) thymine kinase (TK) gene and viral growth factor (VGF) to form a mutant vvDD virus, which replicated only in cells with mutated RAS/mitogen-activated protein kinase (MAPK)/ERK signal transduction pathways. CIK cells were infected with vvDD virus to form VVDD-CIK, which did not affect the expression levels of NKG2D, CD3, CD16, or CD56 or the tumor killing activity of CIK cells. This achieved a favorable antitumor effect and improved the survival rate of mice [44].

##### Chimeric Antigen Receptor T (CAR-T) Cells

Recent studies have shown that using CAR T cells to systematically deliver OVs can also achieve better therapeutic results [52]. On the one hand, as previously mentioned, OVs can remodel the local TME and improve T-cell recruitment and effector function. On the other hand, CAR-T cells can help OVs overcome their limited antitumor effect on distant metastases [53,54]. The Ningbo Zheng team reported that oncolytic myxoma virus (MYXV)-infected tumor-specific T (TMYXV) cells, expressing the CAR or the T-cell receptor (TCR), systemically deliver MYXV to solid tumors in order to overcome primary resistance. The tumor eradication by CAR/TCR-TMYXV cells was attributed to the induction of tumor cell autophagy. T-cell-derived IFNγ-AKT signaling synergizes with the MYXV-induced molecular target of rapamycin (mTOR)-T5-SKP1-vacuolar protein sorting 34 (M-T5-SKP-1-VPS34) signaling to trigger robust tumor cell autophagy. Additionally, CAR/TCR-TMYXV-elicited autophagy functions as a potent bystander killing mechanism to restrain antigen escape [52].

Moreover, the Mayo Clinic team successfully intravenously injected oncolytic vesicular stomatitis virus (VSV) and reovirus using dual-specific CAR T cells. This treatment led to the prolonged survival of mice with subcutaneous melanoma and intracranial glioma tumors [55]. Furthermore, building on the first-generation ONCOTECH technology and to further facilitate clinical translation, researchers utilized HEK293 cells (human embryonic kidney cells) expressing specific antigens as production cells. After infecting these production cells with oncolytic viruses, they isolated the microvesicles containing the OVs. These microvesicles, presenting specific antigens on their surfaces, were then used to bind to chimeric antigen receptors on CAR-T cells to form second-generation ONCOTECH technology. This enhanced technology was tested for targeting efficiency and therapeutic efficacy in mice with a human immune system, including models such as human orthotopic brain tumors, human orthotopic lung cancer, and human pancreatic cancer patient-derived xenografts (PDXs). The results demonstrated that ONCOTECH effectively inhibited the growth of various tumor types [40].

#### 2.3.3. Improvements

Although T cells exhibit high efficiency in tumor homing, two important factors need to be considered: the limited proliferation capacity of carrier T cells [56] and the various immunosuppressive mechanisms excluding T lymphocytes in the tumor microenvironment. Erwei Song and colleagues discovered that regulator of G-protein signaling (RGS) family proteins, which are GTPase-activating proteins (GAPs), impair T-cell migration by accelerating the hydrolysis of GTP to GDP on the Gα subunit, thereby inactivating downstream pathways. Additionally, RGS1 inhibits the activation of calcium pathways downstream of GPCRs by binding to the chemokine receptors C-X-C chemokine receptor 3 (CXCR3), CXCR4, and C-C chemokine receptor 4 (CCR4), which, in turn, suppresses the ERK and AKT signaling pathways, ultimately hindering T-cell migration to tumor sites. Consequently, knocking down RGS1 expression can significantly enhance T-cell infiltration into tumors [40]. Recent studies have also suggested that the expression of acidity-related genes in TAMs is negatively correlated with T-cell infiltration scores. Extracellular acidification inhibits the expression of methyltransferase-like 3 (METTL3) and its mediated RNA N6-methyladenosine (m6A) modification, leading to a reduced expression of its downstream target, integrin β1 (ITGB1). This suppression hampers the formation of T-cell pseudopodia, thereby weakening T-cell motility and infiltration. Thus, modulating METTL3 activity or promoting integrin ITGB1 expression could enhance T-cell infiltration into solid tumors [57]. In short, it is important to develop various approaches to improve lymphocyte trafficking into tumors.

### 2.4. Mesenchymal Stem Cells (MSCs)

#### 2.4.1. Background

MSCs, also known as multipotent stromal cells, are a type of pluripotent stem cell originating from the mesoderm and primarily found in connective tissues and organ stroma. They offer five major advantages when used to deliver OVs: (1) MSCs can protect OVs from being recognized and destroyed by the host immune system, thereby enhancing their spread and persistence. On the one hand, in vitro-expanded MSCs do not express HLA class II molecules or co-stimulatory molecules such as CD40, CD80, CD83, CD86, and CD154, thereby blocking the activation of T cells and avoiding immune recognition and attack. On the other hand, MSCs exert immunosuppressive effects through the release of factors such as interleukin-6 (IL-6), IL-10, TGF-β, heme oxygenase-1 (HO-1), indoleamine 2,3-dioxygenase (IDO), and inducible nitric oxide synthase (iNOS). (2) MSCs can not only carry the viruses but can also serve as a site for viral replication. Due to their low immunogenicity and unique microenvironment, MSCs can provide a relatively “safe” niche for viruses, allowing them to replicate without being targeted by the host immune system. (3) MSCs exhibit high tumor-targeting potential. Researchers have indicated that, in the TME, tumor cells and immune cells can release various chemokines and their receptors, which can guide the homing of MSCs to tumors [58,59], such as SDF-1/ CXCR4, hepatocyte growth factor (HGF)/tyrosine-protein kinase Met (c-Met), vascular endothelial growth factor (VEGF)/vascular endothelial growth factor receptor (VEGFR), and monocyte chemoattractant protein 1 (MCP1)/C-C chemokine receptor type 2 (CCR2) [59]. Adhesion molecules also play a crucial role in MSC adherence to the tumor vasculature and subsequent tumor homing. Guided by chemokines, MSCs initially roll in the bloodstream in a selectin-mediated manner, and then they adhere to vascular endothelial cells via integrin-mediated adhesion [60]. These integrins include various adhesion molecules, such as intercellular adhesion molecule-1 and -2, vascular cell adhesion molecule-1, P-selectin adhesion molecule, and integrin β1 [61,62]. Once MSCs attach to the tumor endothelium, the cytokines in TAMs, particularly the pro-inflammatory cytokines, induce the expression of matrix metalloproteinases (MMPs) by the attached MSCs, including MMP-1, MMP-2, MMP-3, MMP-9, and membrane type-1 MMP (MT1-MMP), which are involved in MSC transendothelial migration [63,64] (Figure 2D). However, the mechanisms underlying MSC tumor targeting are not fully understood and require further investigation. (4) MSCs have been shown to possess excellent tumor penetration capabilities and can successfully deliver oncolytic viruses deep into tumor masses. This ability is believed to be closely related to metallopeptidase activity and the high expression of integrin α3 [65]. Furthermore, MSCs’ involvement in TME formation grants them a strong affinity for tumor cells, reducing barriers in the high-pressure tumor stroma [66]. (5) MSCs can remain in tumors for an extended period. Zhang et al. found that MSCs survived in lung metastases for over five days [67]. Li et al. found that MSCs persisted in a mouse glioma model for up to four weeks after intracranial implantation [68].

#### 2.4.2. Applications of MSCs as OV Carriers

Hence, MSCs are particularly attractive as cell carriers for OVs. Tao Jiang’s team created an Adv (Delta-24-RGD) that replicated specifically within tumors and exhibited high affinity; this was achieved by deleting 24 amino acids from the E1A protein and mutating the heparan sulfate-binding protein on the virus surface (amino acid sequence KKT → RGD). Then, bone marrow-derived mesenchymal stem cells (BM-hMSCs) isolated from human bone marrow using bone marrow aspiration were transfected with Delta-24-RGD to carry viral particles. Afterward, these virus-loaded BM-hMSCs were injected into the intracranial tumor model mice via the carotid artery. The results demonstrated that BM-hMSCs loaded with Delta-24-RGD significantly inhibited tumor progression and prolonged the survival of the treated animals [69]. McKenna and colleagues utilized MSCs to systemically deliver a helper-dependent adenovirus (CAd) engineered to express interleukin-12 (IL-12) and a PD-L1 blocking protein. MSCs infected with CAd were able to deliver and release functional viruses to infect and lyse lung tumor cells. Simultaneously, the release of IL-12 and the PD-L1 blocking protein enhanced the antitumor activity of CAR-T cells. This approach, when combined with human epidermal growth factor receptor 2 (HER2)-specific CAR-T-cell therapy, effectively eliminated 3D-cultured tumor spheroids in vitro and inhibited tumor growth in in vivo orthotopic lung cancer models [70]. In addition, MSCs loaded with MYXV [71,72], HSV [73,74], measles virus (MV) [75], and reovirus [76] have been delivered to tumors in mouse models.

#### 2.4.3. Improvements

Although numerous studies have demonstrated that MSCs loaded with OVs have better therapeutic efficacy than bare OVs, there are still some unsatisfactory effects. Several studies have suggested that MSCs play a role in the disincentive immune system. They may be counterproductive to the antitumor immune response, which OV therapy aims to induce [77]. In the innate immune system, tumor-associated mesenchymal stem cells (TA-MSCs) play a role in immune modulation. On the one hand, they secrete prostaglandin E2 (PGE2) and exosomes, promoting the recruitment and differentiation of immunosuppressive M2 macrophages [78]. They also secrete HGF and C-X-C motif chemokine ligand 3 (CXCL3), which recruit myeloid-derived suppressor cells (MDSCs) [79]. On the other hand, TA-MSCs can secrete TGF-β, IL-6, PGE2, and microRNAs, which can downregulate the expression of activating NK cell receptors such as NKp44, NKp30, NKG2D, DNAX accessory molecule-1 (DNAM-1), and NKG2A, leading to impaired NK cell function [80]. Moreover, the IL-6, PGE2, and microRNAs secreted by MSCs can inhibit the differentiation of monocytes into dendritic cells (DCs) and the maturation of immature DCs [81]. Furthermore, MSCs can exert potent immunosuppressive effects on γδ-T cells through the production of PGE2 via a cyclooxygenase-2 (COX2)-dependent mechanism [82]. Therefore, to improve clinical outcomes, a logical next step is to optimize MSCs.

## 3. Binding with Proteins

### 3.1. Virus-Protein Corona Replacement Strategy

Ph.D. Hanwei Huang et al. believed that the key to improving the circulation of OVs is to prevent the formation of the virus-protein corona rather than to simply prevent the binding of neutralizing antibodies or complements to the OVs [83]. Consistent with this conclusion, Kariem Ezzat and his colleagues discovered that respiratory syncytial virus (RSV) interacts with proteins in the host’s biological fluids, forming a “virus-protein corona” on the viral surface. This protein coating enhances the infectivity of the virus and promotes the formation of plaques associated with neurodegenerative diseases, such as Alzheimer’s disease [84]. 

Moreover, when the coronavirus disease 2019 (COVID-19) virus makes contact with mucosal surfaces or enters the host body, it may interact with a variety of dissolved biomolecules, including proteins, lipids, and sugars, as well as other small molecules, such as hormones and metabolites, to form one or more layers of biomolecules. These biomolecular layers, acquired from the surrounding environment, are referred to as the “acquired corona”, also known as the “virus-protein corona.” Scientists believe that the acquired corona may influence the coronavirus’s spread across different tissues, the penetration of biological barriers, biodistribution, and immunomodulatory functions [85]. Hanwei Huang et al. identified the key protein components of the virus-protein corona by quantifying and conducting a proteomics analysis of the protein corona of OVs, PEGylated (polyethylene glycol) liposome-coated OVs, and PEGylated liposomes. To completely prevent the interaction of OVs with these key proteins and the formation of the virus-protein corona, a virus-protein corona replacement strategy using an artificial virus-protein corona was proposed (Figure 3). In this strategy, an artificial virus-protein corona was formed on OVs through the electrostatic adsorption of cationic polyethylenimine (PEI)-modified OVs with serum albumin and stabilized using a PEGylated liposome coating. This strategy dramatically prolonged the circulation time of the OVs by over 30-fold and increased their distribution in tumors by over 10-fold, resulting in superior antitumor efficacy in primary and metastatic tumor models [83]. The protein corona replacement strategy offers a fresh perspective on the intravenous delivery of OVs, shifting the focus of future research from preventing the interaction of OVs with neutralizing antibodies and complement to preventing the interaction of OVs with crucial viral protein corona components in the bloodstream.

### 3.2. Modification of Key Capsid Proteins

One of the major challenges in the targeted systemic delivery of OVs is the presence of IgM and complement factors in the bloodstream. When human species C adenovirus HAdv-C5 is delivered intravenously, HAdv-C5 is rapidly opsonized by IgM antibodies and coagulation factor X, leading to virus sequestration in tissue macrophages [86]. Previous research has shown that IgM can bind to viruses through “unspecific” low-affinity–high-avidity interactions with regularly structured multifunctional proteins on the virion surface. Ritu R. et al. suggested that natural IgM antibodies can bind to the hyper-variable region 1 (HVR1) of the main HAdv-C5 capsid protein hexon [87]. Svetlana Atasheva and colleagues introduced mutations into the IgM binding site of human adenovirus to prevent the virus from being inactivated in the bloodstream or captured by liver macrophages (Figure 2). Additionally, they replaced some of the adenovirus proteins that interacted with human cell integrins with a sequence from another human protein, laminin-α1, to target the virus to tumor cells. This engineered oncolytic adenovirus was named Ad5-3M. Ad5-3M can avoid inactivation and capture when administered systemically, and it does not induce liver toxicity.

The therapeutic efficacy of Ad5-3M has also been validated in a mouse model of metastatic lung cancer [86]. In addition, Luis Alfonso Rojas et al. inserted an albumin-binding domain (ABD) into the hexon, a major adenovirus capsid protein. The ABD-modified adenoviruses could bind to human and mouse albumin and maintain the infectivity and replication capacity in the presence of neutralizing antibodies (Nabs) [88]. Additionally, scientists are also attempting to replace the natural viral envelope glycoprotein with a glycoprotein that exhibits relative resistance to complement [89,90], as well as incorporating complement-inhibitory proteins into viral particles [91] (Figure 3).

## 4. Nanoparticle (NP)-Based Delivery Systems

NP-based delivery systems have been proven to be an important means of cancer therapy. NPs can target solid tumors through active or passive mechanisms. In terms of passive mechanisms, first, growth-induced solid stress causes a reduction in tumor blood flow and the collapse of lymphatic drainage in the core of the tumor, which is usually accompanied by a decrease in extracellular pH and an increase in necrosis. This is because the active proliferation of cancer cells leads to the compression of blood and lymphatic vessels, especially in its inner regions. Additionally, the poorly organized and leaky vasculature network in tumors, in combination with the increased hematocrit and viscosity of tumor blood, also reduces blood flow rates [92]. In addition, tumor vascular endothelial cells are defective, poorly arranged, and permeable to each other [93]. Due to the large fenestrations in the disorganized tumor endothelium, the slowing of blood flow, and the lack of lymphatic drainage, nanoparticles are retained at the tumor site once they enter. This phenomenon is called the “enhanced permeability and retention” (EPR) effect.

Since it was reported in the late 1980s, the EPR effect has been recognized as a major factor in the enrichment of nanoparticles at tumor sites. Recently, however, some scientists have questioned the mechanism by which nanoparticles enter solid tumors. They suggest that transcytosis may be the main mechanism of nanoparticle enrichment at tumor sites. Phagocytosis is an active metabolic process that requires the reorganization of the endothelial cell cytoskeleton and membrane, including the formation of vesicles capable of engulfing nanoparticles, the formation of transmembrane structures called pore walls, or transport through the cytoplasm. Shrey Sindhwani et al. prepared three sizes (15 nm, 50 nm, and 100 nm) of gold nanoparticles (AuNPs), and the vesicles involved in transporting AuNPs were clearly observed in transmission electron microscopy (TEM) images. The AuNPs exhibited significant interactions with tumor vascular endothelial cells and were absorbed by them, providing direct evidence of phagocytic uptake [94].

Active delivery means specifically binding and directing the movement of nanoparticles to the target tumor tissue inside the body [95], including antibodies, PEG chains, polysaccharides, aptamers, peptides, and small molecules with a strong affinity and specificity for receptors and excessive molecules on tumor cells (Figure 1) [96]; the aim of this is to maximize the OV concentrations in cancer tissues with minimal side effects. In most cases, active and passive delivery occur simultaneously and are not contradictory [97]. Alessandra Iscaro et al. encapsulated an oncolytic adenovirus (Ad[I/PPT-E1A]) into CCL2-coated liposomes to exploit the recruitment of CCR2-expressing circulating monocytes into tumors. The intravenous administration of the nanomedicine resulted in a significant reduction in tumor size and pulmonary metastasis in prostate cancer-bearing mice, whereby a 1000-fold less virus was needed when compared to Ad[I/PPT-E1A] alone [98].

### 4.1. Microbial Nanocomposites

With the continuous advancement of gene sequencing technology, researchers have discovered that microbial communities are present in and an integral part of the TME [99,100]. In 2020, Nejman D et al. [101] conducted a systematic study on 1526 samples obtained from seven different types of tumors and their adjacent tissues, confirming the presence of bacteria in various tumors. Compared to healthy tissues, the special conditions in the tumor microenvironment, such as angiogenesis, hypoxia, nutrient reorganization, and immune suppression, made bacteria more likely to accumulate in tumor tissues [102]. Although the specific mechanisms underlying the natural ability of bacteria to specifically colonize tumors need further investigation, previous reports have shown that both obligate and facultative anaerobes can accumulate and proliferate extensively within tumors after systemic administration. Zheng et al. [103] found that, three days after an intravenous injection of S. typhimurium, the number of bacteria at the tumor site reached 10^10^ CFU/g, a quantity 10,000 times higher than that in normal tissues. Some studies have suggested that this phenomenon occurs because large numbers of bacteria colonize areas further from the normal vasculature in tumor tissues, attributing the accumulation to the disorganized vascular system within tumors, leading to hypoxia, thus creating an environment more suitable for anaerobic bacteria [104]. Given the tumor-homing ability and growth suitability of bacteria within tumors, researchers have proposed the concept of “microbial nanocomposites” (Figure 4A). Encapsulating oncolytic Adv into biomineralized calcium phosphate (CaP) camouflaged and pyranose oxidase (P_2_O) engineered bacterial outer membrane vesicles (OMVs) can achieve autophagy cascade-enhanced antitumor immunotherapy. The CaP shell protects the OVs from clearance by the innate immune system after intravenous injection, enhancing intratumoral accumulation. Once inside the tumor cells, the microbial nanocomposite-constructed P_2_O converts endogenous glucose in situ into H_2_O_2_, increasing oxidative stress levels and triggering tumor autophagy. On the one hand, the in situ synthesis of autophagosomes induced by autophagy provides a site for viral replication during Ads infection, increasing the production of viral particles. On the other hand, Ads-activated autophagy may trigger immunogenic cell death, releasing damage-associated molecular patterns and tumor-associated antigens, thereby activating antitumor immune responses [105]. Zhenning Wang et al. [106] combined liposome-coated oncolytic adenoviruses (OAs) with tumor-homing Escherichia coli BL21 (E. coli-lipo-OAs) to enhance cancer immunotherapy.

### 4.2. Biomineralization

Previous research suggests that metal ions can also be used in cancer treatment. A study by Tsvetkov et al. [107], published in Science, proposed a novel form of cell death induced by copper ions, termed “copper death.” Copper death is closely associated with the tricarboxylic acid cycle (TCA), where ferredoxin 1 (FDX1) acts as an upstream regulator of protein lipoic acidification, promoting the lipoic acidification of dihydrolipoamide acetyltransferase (DLAT) and dihydrolipoamide succinyltransferase (DLST). When intracellular copper ions are present in excess, this protein lipoic acidification modification is reduced. Simultaneously, excess Cu^+^ binds to lipoic acidated DLAT, leading to oligomerization and the formation of polymers, which are insoluble and toxic to cells [108]. The oligomerization of DLAT, which constitutes pyruvate dehydrogenase, results in the loss of pyruvate dehydrogenase activity, inhibiting the conversion of pyruvate to acetyl-CoA and ultimately affecting the TCA cycle. Additionally, some research suggests that copper ions can promote the degradation of PD-L1 in tumor cells, although the mechanism remains unclear [109]. Tong Ge et al. [110] developed a multifunctional oncolytic virus (OA@CuMnCs) using bimetallic ions copper and manganese (Figure 4B). These metal cations formed a biomineralization coating on the surface of Adv, effectively preventing Ads from being cleared by the body after intravenous injection, while the copper ions promoted the degradation of PD-L1, further activating the immune system. 

Furthermore, in this study, Mn^2+^ served as an activator of the stimulator of interferon genes (STING) in the innate immune system. Once the cytoplasmic cGAS-STING pathway (cyclic GMP-AMP synthesis-stimulator of interferon genes) was activated, it induced the production of cytokines, such as IFN-γ [111], thereby activating the immune system. However, after accumulating at the tumor site, OA@CuMnCs released Mn^2+^, which can reduce hydrogen peroxide to oxygen. The increased oxygen content enhanced OA’s replication ability, significantly improving the antitumor effect [112].

### 4.3. Cell Membrane Nanovesicles

Extracellular vesicles (EVs) are nano- to micron-sized lipid membrane-bound vesicles that are secreted into the extracellular environment and transport proteins, lipids, and nucleic acids from cell to cell [113]. They are naturally occurring cargo delivery agents with the potential to be used as vehicles for OVs [114,115]. By using a mouse cancer cell line grown in an immunocompetent syngeneic NFκB-luc2 reporter mouse model and a fluorescent dye to track EVs, the biodistribution and effects of EV-forming stimulants carrying OVs on the immune system were assessed. It was found that EVs enhanced the systemic delivery of the OVs, resulting in improved tumor-selective delivery, a peritumoral immune response associated with the targeted delivery of the virus, enhanced immunogenicity, and the infiltration of CD4^+^ and CD8^+^ T cells [114].

Additionally, Peng Lv et al. developed a type of biotechnological cell membrane nanovesicle expressing PD-1 (PD1-BCMNs) and encapsulating Adv, achieving cancer treatment through immunotherapy checkpoint blockade and oncolytic virus therapy within a single nanoparticle. This study demonstrated that PD1-BCMNs can mask the viral epitopes recognized by neutralizing antibodies, thereby protecting OAs from serum neutralization. Moreover, compared to naked OAs, PD1-BCMNs@OA secrete less TNF-α and IL-6, reducing virus-induced inflammation and toxicity [116] (Figure 4C). Additionally, red blood cells (RBCs) possess characteristics such as natural biocompatibility, low immunogenicity, a high loading capacity, and a prolonged circulation time. Therefore, the use of RBCs for OV delivery has yielded unexpected results [117]. Liu et al. developed a therapy for delivering OVs using RBCs (ELeOVt), where OVs were assembled on the surface of red blood cells using cationic polyethyleneimine (PEI) to link the OVs and RBCs through electrostatic interactions. Experimental results showed that ELeOVt significantly extended the circulation time of OVs, increased their pulmonary distribution by more than tenfold, and markedly improved their therapeutic efficacy against lung metastasis while also reducing organ and systemic toxicity [118].

## 5. Conclusions and Future Perspectives

Many researchers are currently dedicated to finding the optimal delivery system for OVs. In this article, we primarily reviewed three OV delivery systems: cell-mediated OV systemic delivery, protein-mediated OV systemic delivery, and nanoparticle-based delivery systems, which had been summarized in Table 1. These delivery systems play varying roles in enhancing antitumor immunity, protecting OVs from clearance by the body, increasing the tumor-targeting specificity of OVs, and promoting viral replication in the hypoxic tumor microenvironment. In particular, the concept of cell-based biological vector-targeted delivery systems has rapidly developed, showing unique advantages in targeted OV delivery to tumors, such as naturally evading immune surveillance and actively targeting tumors. However, the harsh microenvironment of tumor tissue can cause the rapid death and clearance of cell carriers. Studies have shown that most exogenous MSCs infused into the body undergo rapid apoptosis due to the harsh microenvironment of diseased tissues, thus failing to exert long-term therapeutic effects through mechanisms such as cell differentiation and paracrine signaling [119]. Nanoparticle-based delivery systems are the most extensively studied tumor-targeting OV delivery systems, demonstrating promising development prospects in preclinical and clinical studies of various tumors. However, nanoparticle-based delivery systems also face a series of challenges in tumor-targeted therapy, such as the following: (1) Passive tumor targeting mechanisms based on the EPR effect have recently been found to have limited targeting efficiency for some tumor tissues, and they lack effective targeting for metastases. (2) Some nanoparticles are easily phagocytosed and cleared by macrophages, affecting the delivery efficiency of OVs. (3) The formation of a protein corona on nanoparticles in the bloodstream may affect the targeting efficiency of nanoparticles based on active targeting mechanisms. (4) The dense structure and high interstitial pressure in tumor tissues limit the further drug delivery of nanoparticles into the deep layers of the tumor.

To enhance the preclinical research on targeted delivery of OVs, research on cell carriers should focus on genetically modified OVs and engineered tumor-specific cells to improve the efficacy and safety of targeted therapy. For example, engineered carrier cells can be designed to recognize tumor-associated antigens and sense specific microenvironmental changes. Upon detecting tumor signals, these carrier cells can autonomously activate genes that initiate OV replication, thereby preventing premature OV release, which may not only hinder OVs from reaching the intended tumor site but also cause systemic toxicity. Additionally, OVs can be genetically engineered to achieve stable replication under hypoxic conditions, thereby enhancing their survival and replication capacity within tumor tissues. For nanoparticle-based delivery systems, research should prioritize combination strategies with other therapeutic modalities, such as coupling nanoparticles with tumor-homing cells or bacteria to improve their tumor-targeting capability. This approach aims to optimize OV delivery while minimizing off-target effects and systemic toxicity.

The most critical challenge in clinical research on OV delivery strategies is translating preclinical findings into safe clinical trials. Currently, two clinical trials (NCT03896568 and NCT02068794) are underway, both utilizing MSCs as delivery carriers. Successfully translating these therapies into clinical practice depends on overcoming numerous technical, economic, and regulatory challenges. First, systematic optimization of infection parameters—including viral concentration and in vitro incubation time—can enhance viral load and therapeutic outcomes. Second, combining delivery carriers with other treatment modalities may generate synergistic effects that amplify the overall antitumor response. Third, ensuring the stability and consistency of manufacturing processes is a key challenge, directly impacting treatment reliability and efficacy. Finally, comprehensive adverse event monitoring protocols must be implemented to guarantee patient safety throughout the therapy.

## Figures and Tables

**Figure 1 ijms-26-06900-f001:**
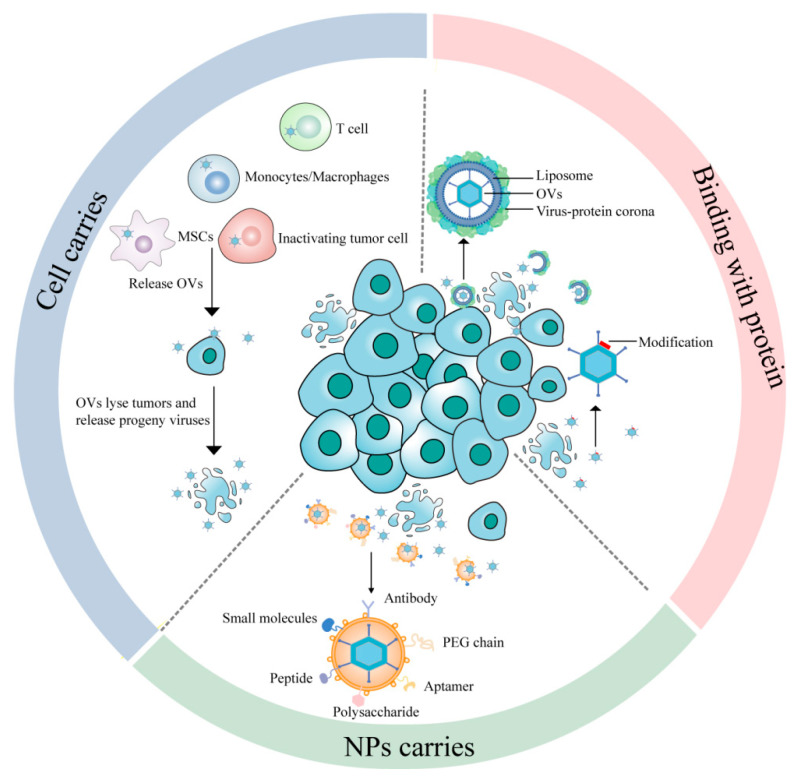
Several effective delivery strategies for OVs, including cell carriers, binding with proteins and nanoparticle (NP) carriers. (1) Cell carriers: utilizing the intrinsic tumor-homing capacity of cellular carriers, including inactivating tumor cells, monocytes/macrophages, T cells and MSCs, to deliver OVs to tumor tissues. Following targeted delivery, OVs infect and lyse tumor cells, releasing progeny viruses; (2) Binding with proteins: the virus-protein corona replacement strategy forms an artificial protein corona on OVs to prevent neutralization by antibodies and complement, while suppressing natural corona formation; The modification of key capsid proteins strategy introduces mutations on the viral capsid surface to evade recognition in the bloodstream; (3) NPs carriers: NPs deliver OVs through active targeting mechanisms. Active delivery means specifically binding and directing the movement of nanoparticles to the target tumor tissue inside the body, including antibodies, PEG chains, polysaccharides, aptamers, peptides, and small molecules with a strong affinity and specificity for receptors and excessive molecules on tumor cells.

**Figure 2 ijms-26-06900-f002:**
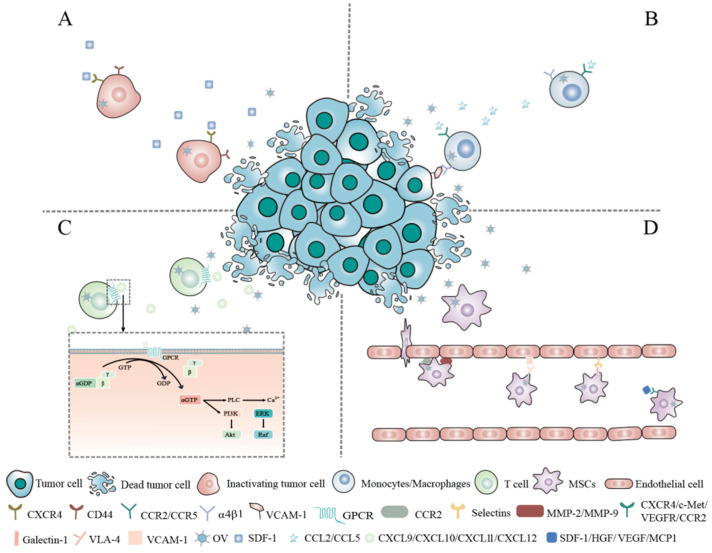
Tumor-homing mechanisms of cell carriers. (**A**) The homing and retention capacity of tumor cells are associated with the expression of CXCR4 and CD44 on the cell membrane; (**B**) Macrophages can specifically recognize VCAM-1 via α4β1 integrin, enabling targeted homing to tumor tissues. Additionally, tumor-secreted CCL2 and CCL5 recruit macrophages into the TME; (**C**) Tumor-secreted chemokines recognize GPCRs on T-cell surfaces and trigger downstream signaling pathways, facilitating the migration of T lymphocytes to the tumor site; (**D**) The molecular mechanism of MSCs homing involves five key steps: (1) Chemokine-receptor interaction; (2) Adhesion molecule-mediated anchoring; (3) Extravasation; (4) Transendothelial migration; (5) Interstitial migration.

**Figure 3 ijms-26-06900-f003:**
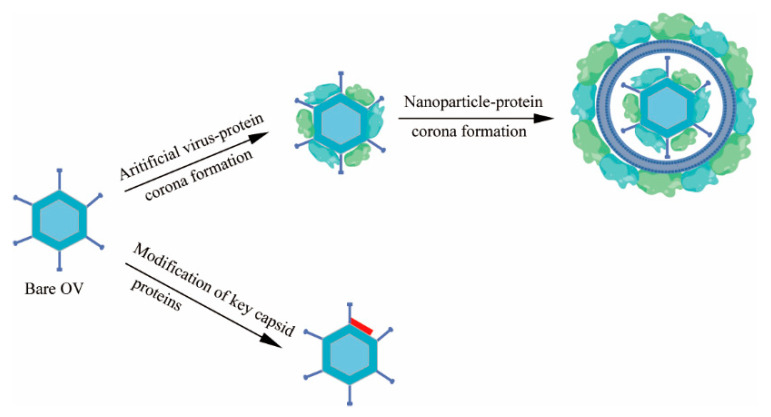
Preparation of binding with protein carriers. Protein corona formation and modification of key capsid proteins.

**Figure 4 ijms-26-06900-f004:**
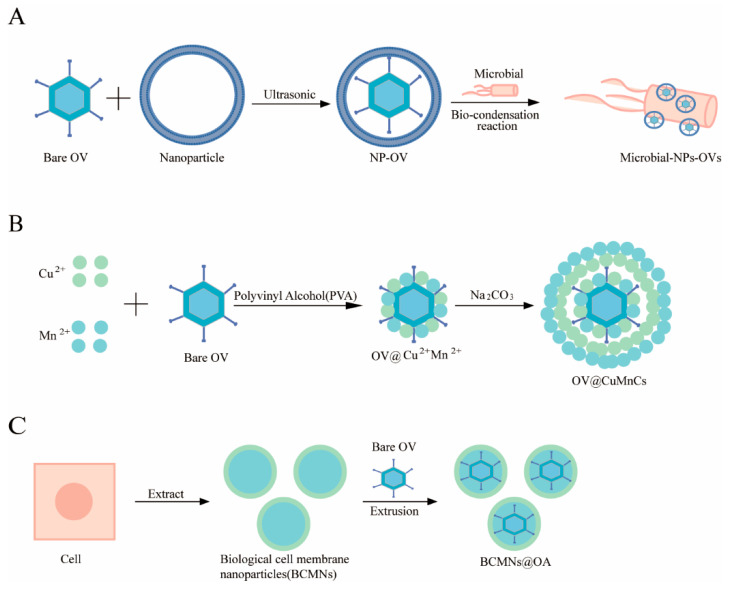
Preparation of NPs carriers. (**A**) The surfaces of OVs were encapsulated by NP layers to retain the activation of OVs during systemic circulation. The “microbial-OV conjugate” was then constructed by combining tumor-homing microbial with NP-OA through a biocondensation reaction, named microbial NPs-OVs; (**B**) OA carries a negative charge at physiological pH because of carboxyl-rich anionic peptides on its surface. Adding MnCl2 and CuCl2 causes Mn^2+^ and Cu^2+^ to adsorb onto OA. Subsequent sodium carbonate addition under physiological conditions stops ion polymerization, spontaneously forming a moderately thick mineral shell around OA; (**C**) Preparation of BCMNs@OA.

**Table 1 ijms-26-06900-t001:** Key features of each delivery strategy.

Delivery System	Tumor Targeting	Delivery Efficiency	Immune Escape	Antitumor Effect In Vivo	References	
Cell-based OV delivery	Tumor cells	CD44 and E-cadherin on TC-1 mouse lung cancer cells.	More than 110-fold enrichment of LNT-Ad11.	83% of OVs were protected from antibody neutralization.	The number of M2 cells and Treg cells was reduced; CD4^+^ and CD8^+^ and T cells priming is enhanced.	[28]	
Monocytes/Macrophages	Binding to CCL2 in TME via CCR2 on the cell surface.	500 out of the 10^6^ intravenously injected cells accumulated in tumors.	\	Tumor-bearing mice exhibited a 1.5-fold increase in overall survival.	[35]	
T lymphocytes	Chemokines, such as CXCL9, CXCL19 ect, secreted by the tumor promoted T-cell migration into tumors.	The delivery of the OVs–T cell chimera (ONCOTECH) achieved a 3.8-fold increase in viral accumulation.	B16OVA cell membrane with MHC-I–OVA could protect eOA from neutralization by anti-Ad5 antibody.	A single administration of ONCOTECH resulted in an 80% survival rate over 70 days.	[48]	
MSCs	Chemokines, such as SDF-1, HGF ect, secreted by the tumor recruited MSCs into tumors.	Shielded MYXV accumulated in lung tumors at 2 h after post-IV injection.	\	The number of pulmonary foci was reduced twofold.	[72]	
Binding with proteins	Virus-Protein Corona Replacement Strategy	DSPE-Poly(2-ethyl-2-oxazoline) (DSPE-PEOZ), a pH-sensitive phospholipid, enabled viral release in the acidic TME.	Increased OVs loading in tumors by more than 10-fold.	Prolonged the circulation time of OVs by more than 30-fold.	Tumor growth was suppressed by more than sevenfold.	[83]	
Modification of key capsid proteins	The HAdv penton base RGD loop interacted with cellular integrins of αvβ3 andαvβ5, promoting efficient virus infection.	Ad5-3M was detected in tumors 12 days after post-IV injection and persisted for 80 days.	Modified Ad5-3M avoided sequestration in liver tissue and poorly activated inflammatory cytokines.	Tumor-bearing mice exhibited a 6.6-fold increase in overall survival.	[86]	
Nanoparticle (NP)-based delivery systems	Microbial Nanocomposites	Hypoxia, aberrant tumor vasculature, and the immunosuppressive TME promoted bacterial self-colonization in tumors.	Self-propelled E. coli-lipo-OAs demonstrated over 170-fold enrichment in tumors.	Liposomes possessed both bioprotective and biocompatible properties.	E. coli-lipo-OAs significantly promoted dendritic cell (DC) activation and induced long-term immune memory in mice.	[106]	
Biomineralization	Mn^2+^ achieved tumor-targeted accumulation via pH-gating effects in acidic TME.	At 48 h after post-IV injection, OA@CuMnCs showed over 50-fold higher viral titers in tumors.	Metal cations formed a biomineralized Adv coating to prevent immune clearance.	OA@CuMnCs enhanced T-cell infiltration and converted “cold” tumors to “hot” tumors.	[110]	
Cell Membrane Nanovesicles	PD-1-engineered biomimetic nanovesicles targeted tumor tissues by binding to PD-L1 on tumor cell surfaces.	PD1-BCMN@OA showed approximately twofold higher accumulation in the tumor than naked virus.	PD1-BCMN (bioengineered cell membrane nanovesicles with PD-1) masked viral epitopes recognized by neutralizing antibodies.	PD1-BCMNs could bind to PD-L1 to activate TILs and elicit a strong antitumor immune response.	[116]

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
