# Peer review of "Systemic Delivery Strategies for Oncolytic Viruses: Advancing Targeted and Efficient Tumor Therapy"

_ijms, 2025, doi:10.3390/ijms26146900_

Round 1

Reviewer 1 Report

Comments and Suggestions for Authors

This review summarizes a range of delivery strategies for oncolytic viruses, including cell-based systems, protein modifications, nanoparticles, and microbial platforms. The authors have made a commendable effort to include recent advances. However, the manuscript lacks higher-level synthesis and cross-cutting insights across these approaches. Significant revision is needed to improve clarity, structure, and conceptual integration before it can be considered for publication.

  1. The current structure treats each delivery method independently. It would be helpful to introduce a summary table that compares key features, such as delivery efficiency, immune evasion, and tumor targeting for each strategy. Defining a set of evaluation criteria could also guide future development in the field.
  2. Many approaches remain at the preclinical stage. The manuscript should better highlight which strategies have advanced into clinical trials and discuss the practical barriers for clinical translation (e.g., immune toxicity, scalability, regulatory approval).
  3. Mechanisms are discussed in multiple places, sometimes with overlap. Consider integrating these insights into a unified figure or schematic that illustrates how different methods overcome delivery challenges.
  4. Some sections, such as the one on T lymphocytes are overly dense. Breaking them into smaller, focused parts with consistent structure (e.g., background, example studies, advantages/limitations) would improve readability.
  5. The final section should go beyond listing challenges. A forward-looking perspective such as the potential for combining strategies or developing standard evaluation models would strengthen the paper’s impact.
  6. There are numerous grammatical issues and overly long sentences throughout the text. The authors are encouraged to seek assistance from a native English speaker or professional editing service.

Author Response

Dear editor,

Greeting for you. Thank you very much for your help. Now we have revised our manuscript according to these comments. Please review it again. Thank you much. We have revised the entire manuscript and had it polished by English language professionals. Additionally, we have re-edited the figures to make their meanings more clearly presented.

Best wishes!

Ph.D. Ou Xia

Reviewer 2 Report

Comments and Suggestions for Authors

In this manuscript, the authors reviewed several delivery methods for oncolytic viruses targeting tumor cells including cells-mediated OVs systemic delivery, binding with protein, NPs carries et al. Thus, it is novel for the readers. However, there are some questions for this manuscript. 

1. The title of this manuscript needs to be modified. This manuscript mainly discusses the improved methods and strategies of systemic delivery for tumor treatment with oncolytic viruses. Therefore, the title needs to emphasize these points.
2. There is a lack of highlighted picture in this manuscript.
3. The legend in Figure 1 shows the Chinese characters "、". The legend description should be more detailed and consistent with the content of the picture, including the introduction of cell types, modification methods and processes, etc.
4. The logical arrangement of the manuscript needs to be improved. For example, Section 2 is "Cell carries", and then sections 3-6 describe different cell vectors and should belong to Section 2, such as 2.1- 2.4. Other parts of the full text should also be checked throughout the manuscript.
5. There are grammatical errors in the manuscript, which should be carefully checked.

Author Response

(The authors gave the same response as above.)

Round 2

Reviewer 1 Report

Comments and Suggestions for Authors

I have carefully examined the authors’ responses and the updated version of the paper. The revisions have substantially improved the overall clarity, structure, and scientific value of the manuscript. I am pleased to recommend it for acceptance in its current form.

Reviewer 2 Report

Comments and Suggestions for Authors

All questions have been issused. I have no more questions.